# Morphological and Molecular Investigation of Non-*Simulium damnosum* Black Flies in Cameroon Using Nuclear ITS 2 and Mitochondrial Cox 1 Genes

**DOI:** 10.3390/insects16060572

**Published:** 2025-05-28

**Authors:** Pierre Kamtsap, Archile Paguem, Flore Nguemaïm Ngoufo, Alfons Renz

**Affiliations:** 1Institute for Evolution and Ecology, Department of Comparative Zoology, University of Tübingen, Auf der Morgenstelle 28, 72076 Tübingen, Germany; alfons.renz@uni-tuebingen.de; 2Onchocerciasis Program Field Station of the University of Tübingen, Ngaoundéré, Cameroon; achillepaguem@yahoo.fr; 3Faculty of Agriculture and Veterinary Medicine, Department Veterinary Medicine, University of Buea, Buea P.O. Box 63, South West, Cameroon; 4Faculty of Health Sciences, University of Bamenda, Bambili Po. Box 39, Cameroon; ngflorema@yahoo.fr

**Keywords:** *Simuliidae*, Cameroon, molecular, identification, black fly, *vorax*, Menchum, *dentulosum*, Soramboum, Mawong

## Abstract

Using morphological and molecular techniques, we investigated the biodiversity of black flies in Cameroon, with a special emphasis on non-*Simulium*-*damnosum* species. After gathering 1184 pupae from 13 different locations, we used gill morphology and DNA sequencing (Cox1 and ITS2 genes) to identify 19 species. The first identification of 2 undescribed (based on identification keys we used) *Simulium* species in Cameroon and the validation of the known onchocerciasis vectors, *Simulium vorax* (first time to be described in Cameroon) and *Simulium dentulosum*, are important discoveries. For reference, DNA sequences were uploaded to GenBank. This study emphasizes the advantages of molecular approaches in revealing the diversity of cryptic species and the drawbacks of conventional morphological techniques. The most widely dispersed species was found to be *Simulium cervicornutum*, while species such as *S. alcocki* and *S. kenyae* showed restricted distributions. This study highlights the possibility that, in the right circumstances, non-*damnosum* species could spread illness, urging increased molecular analysis and vector surveillance in Cameroon. To improve vector control techniques and obtain a deeper understanding of species-specific roles in pathogen transmission, future research should integrate whole-genome sequencing and more comprehensive ecological and taxonomy studies.

## 1. Introduction

Despite the importance of black flies in the transmission of various parasites, few studies have been carried out concerning the general diversity and identification (on the molecular level) of black flies in Cameroon. Larvae and pupae are widespread in fast-flowing water of rivers and tributaries [1,2]. Some adult females are bloodsucking, their life cycle includes feeding on vertebrates, e.g., wild and domestic animals and humans, as blood-hosts [3,4]. Thereby, black flies transmit important pathogens such as *Onchocerca volvulus* and *O. ochengi*, causative agents of human onchocerciasis (river blindness) and bovine onchocercosis [5,6], respectively, *Leucocytozoon* to birds in Asia and North America [7,8], and numerous other filarial parasites of wild and domestic animals (*Onchocerca dukei*, *O. ramachandrini*, *O. lienalis*, *O. lupi*, *O. flexuosa*, *Lappnema* sp.) [3,4,9,10,11,12].

Black flies (Diptera: Simuliidae) are present worldwide, there being 31 genera containing 2348 species (2331 living and 17 fossil species) [13]. In Africa, 124 black fly species have been described, mostly in the Ethiopian region in the 1950s [14]. In Cameroon, 55 species have been morphologically described (Table 1) [13] and only a few are as yet well classified [15]. Similarly, species diversity and identity on the molecular level in Cameroon remain insufficiently understood. Not only knowledge of the geographical characteristics and the geographical diversity of the members of the *S. damnosum* complex but also the determination of the differentiation scale and estimation of the distance between populations are necessary for any planned vector control.

In a previous study [16], we focused on the molecular diversity of members of the *Simulium damnosum* complex in Cameroon, which are the main local vectors of *Onchocerca volvulus*, *O. ochengi*, and *O. ramachandrini*. We now extend this study to the *non-S. damnosum* black flies, which constitute the majority of all species in this country.

A wide range of cytological and molecular markers have been used for population studies in Simuliidae [17,18]. These include chromosomal inversions [19], allozymes [20], and random amplified DNA polymorphisms (RAPD) [21]. Furthermore, the sequencing of mitochondrial Cytochrome oxidase 1 (Cox1) genes [22,23], nuclear genes (ITS), and microsatellite loci analyses [23] have been carried out.

The ITS2 region of nuclear ribosomal DNA is regarded as one of the candidate DNA barcodes because it possesses a number of valuable characteristics, such as the availability of conserved regions for designing universal primers, the ease of its amplification, and sufficient variability to distinguish even closely related species [24].

All the above-mentioned techniques have led to the conclusion that morphological classifications do not distinguish between many populations that should be recognized as true species (‘cytospecies’, etc.). An ideal barcode should be sufficiently variable to identify closely related species, while carefully identifying distantly related species. Indeed, a prediction has been made that, worldwide, more than 3000 black fly species are potentially undiscovered morpho species and sibling species [2].

Because of their impact on public and animal health, the correct identification of this insect group is of a fundamental importance in order to provide correct information on species distribution and biology so that targeted control measures can be correctly applied. However, standard methods for black fly species identification are mainly based on morphology, which typically requires expert knowledge, and sometimes the resolution can be poor because of the presence of hidden diversity [25,26,27].

In the present study, we used the morphological aspect of pupal gills and developed a molecular platform based on the ITS2 and the Cox1 in order to support the species identification of the poorly studied black fly fauna of Cameroon.

## 2. Materials and Methods

### 2.1. Source of Material and Morphological Identification

Substrates to which pupae were attached were collected by hand in Cameroon (Figure 1, Table 2) and included trailing vegetation, debris, stones, and refuse such as plastic and glass. Pupae which were attached to their substrate were immediately placed in boxes and covered with wet tissue. Pupae were removed from substrates, cleaned using a fine brush and forceps, and preserved in 70 to 96% ethanol.

Pupae were identified under a Wild M5 dissection and a Zeiss Axioplan compound microscope by using standard keys as described by Freeman and de Meillon and others (molecular aspect) [14,28,29]. Identified species were cleared in 10% potassium hydroxide (KOH) solution for about 24 h at room temperature. Gills were cut out carefully using fine needles and forceps, transferred on a clean slide containing a drop (approximately 50 μL) of polyvinyl lactophenol, and covered with a coverslip. All mounted slides were kept on a heat bloc (Omni lab Jürgens, Germany) set at 60 °C for approximately 24 h. Images were taken with an incorporated Canon EOS-650D camera.

### 2.2. DNA Extraction, PCR, and Sequencing

The Wizard Genomic DNA Purification Kit (Promega, Madison, WI, USA) [30] was used, as instructed by the manufacturer, to extract total genomic deoxyribonucleic acid (DNA) from the individual pupae that had been identified morphologically [28]. Gene amplification of the mitochondrial protein-coding gene CoxI, which is about 650 bp long, was performed by the Polymerase Chain Reaction (PCR) with the Lep primers: forward (5′-ATTCAACCAATCATAAAGATATTGG-3′) and reverse (5′-TAAACTTCTGGATGTCCAAAAAATCA-3′) [21], whereas ITS2, which is about 400 bp long, was identified by using forward (5′-TGTGAACTGCAGGACACAT-3′) and reverse (5′-ATGCTTAAATTTAGGGGGT-3′) primers [31,32]. All the PCRs were performed in a final volume of 25 µL comprising 2 µL genomic DNA, 5 µL Promega 5× DNAgo Buffer, 2 mM MgCl_2_, 0.25 mM each dNTPs, 50 pmol forward and reverse primers and 1 U Promega Taq Polymerase (Promega) [30]. Amplifications were performed in a Master Cycler (Eppendorf Master Cycler). For the Lep primers, PCR consisted of an initial denaturation (95 °C, 2 min), followed by 35 cycles of denaturation at 95 °C for 30 s, an annealing step at 51 °C for 30 s, an extension at 72 °C for 60 s, and then a final extension at 72 °C for 5 min [30]. For the ITS2, PCR consisted of an initial denaturation at 94 °C for 2 min, followed by 35 cycles of denaturation at 94 °C for 40 s, an annealing step at 51 °C for 60 s, extension at 72 °C for 60 s, and then a final extension at 72 °C, for 5 min [32]. The amplified amplicons were checked by electrophoresis on a 1.5% agarose gel. Finally, PCR products were sent to a commercial sequencing facility (Macrogen, Amsterdam, The Netherlands).

### 2.3. Sequence Analysis

All bi-directional sequences were combined to produce a single consensus sequence in Geneious Prime v. 2023.2.1. The alignment was performed with ClustalW with default parameters, and the neighbor-joining (NJ) analysis was undertaken using the K2P distance to represent species distribution patterns in the NJ tree. The robustness of the NJ tree was calculated using the bootstrap methodology employing 1000 as pseudoreplicates. All obtained sequences for Simuliidae from this study (GenBank accessions for ITS2: see Appendix A) were chosen to encompass the range of Simuliidae species occurring in Cameroon based on morphology. The optimal tree with the sum of branch length is shown. The trees are drawn to scale, with branch lengths in the same units as those of the evolutionary distances used to infer the phylogenetic tree. The ME tree was searched using the Close Neighbor Interchange (CNI) algorithm at a search level of 1. The number of nucleotide sequences implicated in the analysis is indicated. Codon positions included were 1st + 2nd + 3rd + Non-coding. All ambiguous positions were removed for each sequence pair. The number of the total position in the final dataset is indicated. We analyzed the dataset in MEGA v.7 [33].

## 3. Results

### 3.1. Morphological Identification

Thirteen sample collection points (see Table 2 with collection sites, sample collector, and collection date; Appendix A with the number of each species per location and percentage of appearance of each species from a specific location) were included in this study. A total of 19 non-*Simulium damnosum* species were identified by morphology based on pupae respiratory gills including *Simulium dentulosum* type A (14 filaments per respiratory gill), *dentulosum* type B (16 filaments per respiratory gill), *dentulosum* type C (12 filaments per respiratory gill); *Simulium adersi*; *Simulium alcocki*; *Simulium bovis*; *Simulium cervicornutum*; *Simulium medusaeforme* f. Pomeroy; *S. medusaeforme* f. *hargreavesi*; *Simulium hirsutum*; *Simulium katangae*; *Simulium kenyae*; *Simulium nigritarsis*; *Simulium ruficorne*; *Simulium schoutedeni*; *Simulium unicornutum*; *Simulium vorax*; *and* 2 not yet described *Simulium* species based on the identification keys that we used (*Simulium* undescribed 1 and 2) see Figure 2a–s.

The heatmap (Figure 3) and the Appendix A reveal strong spatial patterns in species dominance and diversity, with *S. cervicornutum* emerging as the most widely distributed and numerically dominant species across multiple locations, including Mawong River, Menchum Falls, and around IRAD. Conversely, *S. medusaeforme* f. *hargreavesi* demonstrates extreme localization, dominating Soramboum (96.97%). Species like *S. katangae* and the undescribed species exhibit regional importance, particularly around IRAD and Menchum Falls. High-diversity areas such as Mawong River, despite being dominated by *S. cervicornutum*, host over 12 species. In contrast, sites like Karna Manga and Aladji Marafat show near-monospecific populations.

### 3.2. Molecular Identification

A total of 1184 non-*Simulium* black fly specimens were collected from 14 distinct geographic locations in the Ethiopian region, encompassing varied ecological zones. Genomic DNA was successfully extracted from and subjected to PCR amplification targeting the nuclear ITS2 region and the mitochondrial Cox1 gene.

Sequences were critically assessed for quality upon receipt from the sequencing facility. Only high-quality sequences with clear chromatograms were included in subsequent analyses. Despite genetic variation, individuals belonging to the same species consistently clustered together regardless of their ecological zone, as illustrated in Figure 4.

PCR amplification of the ITS2 region was successful, producing clear, single bands of approximately 400 bp, as visualized by agarose gel electrophoresis. The absence of non-specific products or primer-dimers confirmed the high quality of both the extracted DNA and the primers. Sanger sequencing of these amplicons yielded high-quality bidirectional chromatograms, with average scores exceeding 30 across both strands, ensuring reliable base calling. Sequence alignment revealed conserved regions with minimal variation, supporting the intra-species consistency of ITS2. BLASTn (Figure 4) analysis against GenBank showed >98% similarity to known non-*Simulium* black fly species, thereby confirming species identity. Notably, pairwise sequence divergence ranged from 1.2% to 4.8%, indicating the presence of distinct haplotypes and potential cryptic diversity. Geographic analysis suggested clustering of similar ITS2 sequences within specific regions, hinting at possible population structuring.

To assess broader genetic diversity and phylogeographic relationships, samples were sequenced for the Cox1 gene (~650 bp). A Neighbor-Joining phylogenetic tree based on Cox1 sequences (Figure 4b) resolved the samples into six well-supported clades with bootstrap values >70%. Reference sequences from GenBank aided in confirming the taxonomic identity of each clade. Sequences such as *S. vorax* (MT323206) and *S. ruficorne* (KY421710, unpublished) showed notable alignment, supporting the validity of the identified groupings.

Sequences generated in this study have been deposited in GenBank, with accession numbers provided in Appendix A.

Table 3 presents the estimation of Average Evolutionary Divergence over Sequence Pairs within Groups by ITS2 region. The ITS2 region analysis revealed varying levels of intra-species divergence among sampled *Simulium* species. Very low divergence values (≤0.005) were observed in *S. ruficorne* (0.002), *S. undescribed* 1 (0.003), *S. hargreavesi* (0.004), and *S. alcocki* (0.005), indicating high genetic similarity likely due to recent common ancestry or limited geographic separation. Moderate divergence was found in *S. unicornutum* (0.008) and *S. dentulosum* (0.024), suggesting modest genetic variation, with *S. dentulosum* possibly reflecting population sub-structuring or ecological adaptation. High divergence values (≥0.05) in *S. cervicornutum* (0.053) and *S. katangae* (0.063) point to significant genetic variability, potentially due to cryptic speciation, long-term isolation, or misclassification, warranting further study. *S. nigritarsis* exhibited zero divergence (0.000), indicating identical sequences across individuals, possibly from recent divergence or limited sampling. Divergence for the outgroup could not be calculated, which is expected and does not impact intra-species comparisons.

Table 4 represents the substitution matrix where each entry is the probability of substitution (*r*) from one base (row) to another base (column). Transitions (purine↔purine: A↔G; pyrimidine↔pyrimidine: C↔T) exhibit significantly higher substitution rates (15.5722) compared to transversions (purine↔pyrimidine: A↔T, A↔C, G↔T, G↔C), which have lower rates (4.7139). This pronounced transition bias, a common feature in molecular evolution, is due to biochemical constraints, as transitions involve simpler molecular changes between structurally similar bases and thus occur more frequently. The substitution matrix is symmetric, indicating that the rates of substitution are equal in both directions (e.g., A→G equals G→A), reflecting reversible mutation processes typical of models like Kimura 2-parameter (K2P) and General Time Reversible (GTR). The rates are relative and scaled so that the average substitution rate across all pairs equals one, with the highest rate (15.5722) being approximately 3.3 times greater than the lowest (4.7139), signifying that some substitutions are markedly more probable. This pattern, especially the elevated A↔G and C↔T rates, is characteristic of the ITS2 region, which, while moderately conserved, allows sufficient variability to reveal meaningful substitution trends. Such biases can influence phylogenetic tree topology and affect divergence time estimates, underscoring the importance of accurate substitution rate modeling. Maximum Likelihood Estimation ensures that the matrix best fits the sequence data, enhancing the reliability of phylogenetic inference, molecular clock calibration, and insights into evolutionary pressures acting on specific gene regions. 

## 4. Discussion

The molecular analysis presented in Figure 4a confirms the species identity of non-*Simulium damnosum* specimens through ITS2-based Sanger sequencing. Clear and specific PCR amplification products, alongside high-quality sequence data, demonstrate successful differentiation of multiple non-*Simulium damnosum* taxa. These results support the utility of ITS2 as a reliable marker for species-level resolution within *Diptera*, particularly in *Simuliidae* and related groups.

Our findings align with previous studies that underscore the robustness of the ITS2 region in resolving species boundaries among hematophagous insects [34,35]. The distinct ITS2 sequence profiles observed suggest notable genetic diversity, potentially indicating cryptic species or substantial intraspecific variation—both critical considerations for accurate vector surveillance and ecological research.

Importantly, the analyzed samples originated from diverse geographic regions, and the sequence divergence observed corresponds with known patterns of geographical structuring in black fly populations [36,37]. This geographic differentiation likely reflects local adaptation or historical biogeographic separation and underscores the importance of integrating molecular tools into vector control initiatives.

The identification of non-*Simulium damnosum* species contributes significantly to understanding species composition in black fly communities, with direct implications for vectorial capacity and potential disease transmission. This is especially pertinent in areas where onchocerciasis transmission dynamics may be influenced by non-primary vectors, as recent evidence suggests [38,39].

Our application of both ITS2 and Cox1 molecular markers facilitated precise species identification, revealed intra- and inter-specific genetic diversity, and elucidated geographic variation among non-*Simulium damnosum* black flies. ITS2-based Sanger sequencing proved reliable for routine species confirmation, especially in field settings where morphological identification is complicated by cryptic species or degraded specimens. ITS2 sequences displayed minimal intraspecific variation, aligning with its conserved nature, yet provided high-confidence species confirmation via BLAST matching [34,35].

Conversely, the mitochondrial Cox1 gene—owing to its higher mutation rate—revealed significant haplotype diversity and geographic clustering (Figure 4b), offering insights into population structure and phylogeography. These patterns support previous work using Cox1 barcoding to differentiate black fly populations across regions [36,37]. The formation of geographically distinct clades suggests restricted gene flow due to environmental barriers, breeding site isolation, or host specificity. Understanding this population structuring is crucial, as it may influence vector competence, transmission dynamics, and response to control efforts. Notably, even non-primary vectors could contribute to disease transmission, particularly where zoonotic *Onchocerca* spp. have been detected in non-*Simulium* species [38].

Our findings complement and refine the classical morphological taxonomy established by Freeman and de Meillon for *Simuliidae* in Africa [39]. The molecular approaches utilized here provide enhanced resolution and accuracy in species identification, vital for epidemiological monitoring and biodiversity assessments. Additionally, Sanger sequencing remains a cost-effective and accessible method for field laboratories, enabling rapid and reliable surveillance, particularly in resource-limited settings.

In this study, we have, for the first time, undertaken a combined morphological and molecular analysis of non-*Simulium damnosum* black fly populations in Cameroon. Notably, 2 undescribed *Simulium* species were detected for the first time in Cameroon. The presence of such species in distinct locales suggests that ecological or environmental factors significantly influence their distribution. High species concentration in certain areas may guide conservation and surveillance efforts.

Ecological patterns also emerged: *S. cervicornutum* was found across multiple locations, indicating ecological adaptability. In contrast, species like *S. medusaeforme f. Pomeroy* and *S. vorax* were limited to specific environments (Aladji Marafat), suggesting niche specialization. Similarly, *S. kenyae* and *S. alcocki* were restricted to certain locations, implying highly specialized environmental preferences.

Furthermore, our molecular analysis revealed the presence of *S. vorax* in Mayo Djouroum near Galim, Adamaoua Region. This species clustered with a previously identified *S. vorax* specimen from the Kakoi–Koda focus in the Democratic Republic of Congo, where it has been implicated in onchocerciasis transmission [40]. This suggests that *S. vorax* in northern Cameroon may also play a role in transmission, possibly including novel or zoonotic *Onchocerca* spp.

Additionally, we present molecular data for three subspecies of *S. dentulosum*, which cluster closely together. This species was previously identified as a primary vector in Kakoi–Koda [40]. For the first time, such molecular data for the Cameroonian subspecies of S. dentulosum have been presented. 

Historically, studies in Cameroon have focused on the *S. damnosum* complex as the sole vectors of onchocerciasis [16,41]. However, our results suggest that other blackfly species may also serve as potential vectors under favorable environmental conditions, emphasizing the need to broaden the scope of vector surveillance.

Lastly, our data also identified *S. ruficorne*, previously described in a phylogenetic study of *Simulium* on Réunion Island [42], underscoring the broader biogeographic connections and potential for species migration or introduction.

## 5. Conclusions

According to these data, some species (like *S. cervicornutum* and *S. katangae*) exhibit a large, extensive range, whereas others (like *S. alcocki*) are more location-specific. The diverse distributions of species and their concentration in particular regions imply that ecological elements, including competition, habitat type, and environmental circumstances, are important determinants of species success. To further understand the underlying reasons for these trends and to improve conservation initiatives, more ecological research would be helpful, including habitat surveys and environmental monitoring. Our study should help to provide useful methods or techniques for vector sibling identification and the spatial distribution of sub-Saharan and tropical elements. Nevertheless, other members of the *Simulium*, and not only the *damnosum* complex, need to be included, such as populations from other regions of Cameroon. In addition, analyses of the whole genome might help to improve the resolution of the relationships. Indeed, the application of identification to other African *Simulium* species with a similar geographical distribution might be useful for underlining Simuliid biogeography and evolution in Africa. The implementation of the suggested ideas and methods will aid the planning of undescribed *Simulium* species that could be implicated in the transmission of onchocerciasis and some unknown pathogens that could be transmitted by black flies. Most species in this study showed low to moderate intra-group divergence, supporting the utility of the ITS2 region for species identification and taxonomic resolution among non-*Simulium damnosum* black flies. The elevated divergence in *S. katangae* and *S. cervicornutum* suggests potential taxonomic complexity and the presence of cryptic species. These results provide valuable insights for phylogenetic analysis, species delimitation, and vector control strategies, where accurate identification is crucial.

## Figures and Tables

**Figure 1 insects-16-00572-f001:**
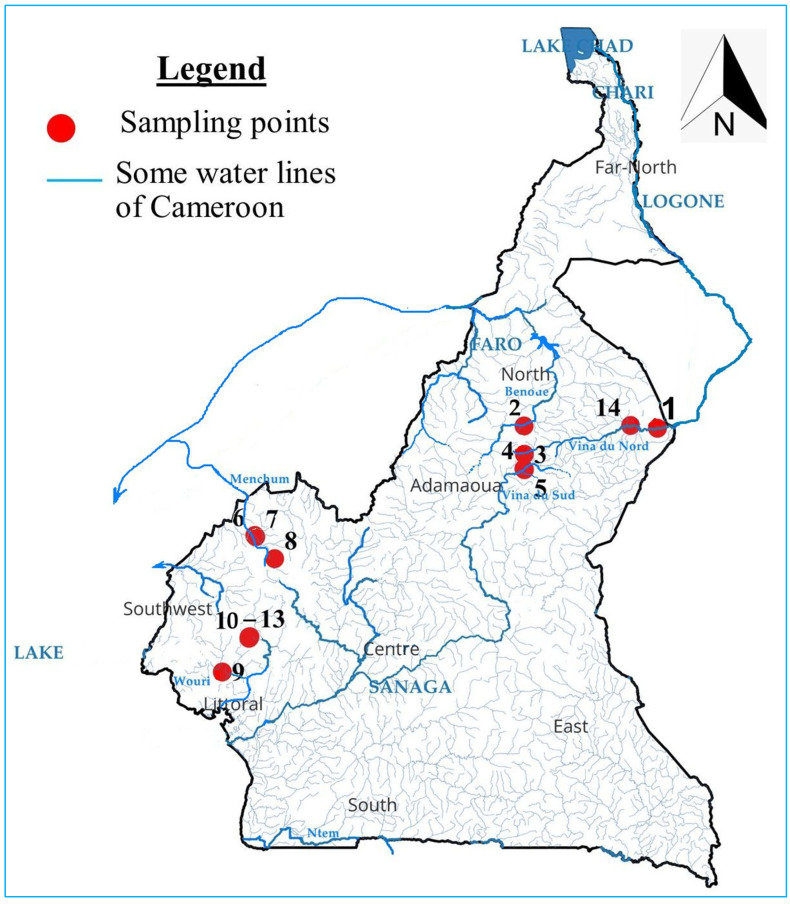
Map of Cameroon showing the localities where samples were collected in this study. Sample points are indicated by red dots with numbers from 1 to 13 (see Table 2 for geographical coordinates).

**Figure 2 insects-16-00572-f002:**
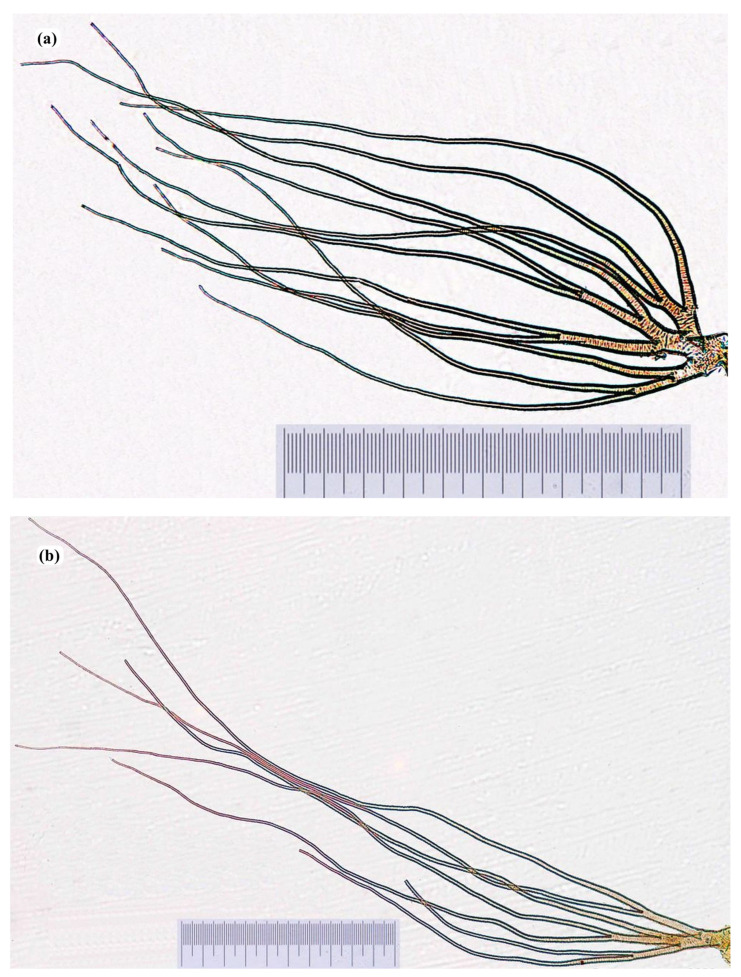
Respiratory gills of pupal stages: (**a**) *Simulium adersi*; (**b**) *Simulium alcocki*; (**c**) *Simulium bovis*; (**d**) *Simulium cervicornutum*; (**e**) *Simulium dentulosum* A; (**f**) *Simulium dentulosum* B; (**g**) *Simulium dentulosum* C; (**h**) *Simulium medusaeforme* f. Pomeroy; (**i**) *S. medusaeforme* forme *hargreavesi* (**j**) *Simulium* undescribed 1; (**k**) *Simulium* undescribed 2; (**l**) *Simulium hirsutum*; (**m**) *Simulium katangae*; (**n**) *Simulium kenyae*; (**o**) *Simulium nigritarsis*; (**p**) *Simulium ruficorne*; (**q**) *Simulium schoutedeni*; (**r**) *Simulium unicornutum*; (**s**) *Simulium vorax*. Scale- 1 mm.

**Figure 3 insects-16-00572-f003:**
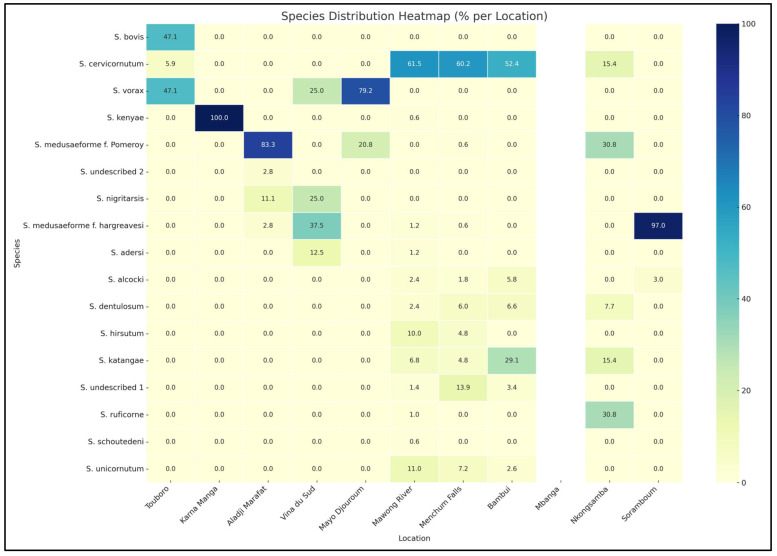
Heatmap showing the percentage distribution of non *Simulium damnosum* species across surveyed locations in Cameroon. Color intensity reflects relative abundance, with dominant species like *S. cervicornutum* and *S. medusaeforme f. hargreavesi* clearly concentrated in specific areas (e.g., Mawong River, Soramboum), while other species show localized or low-frequency patterns.

**Figure 4 insects-16-00572-f004:**
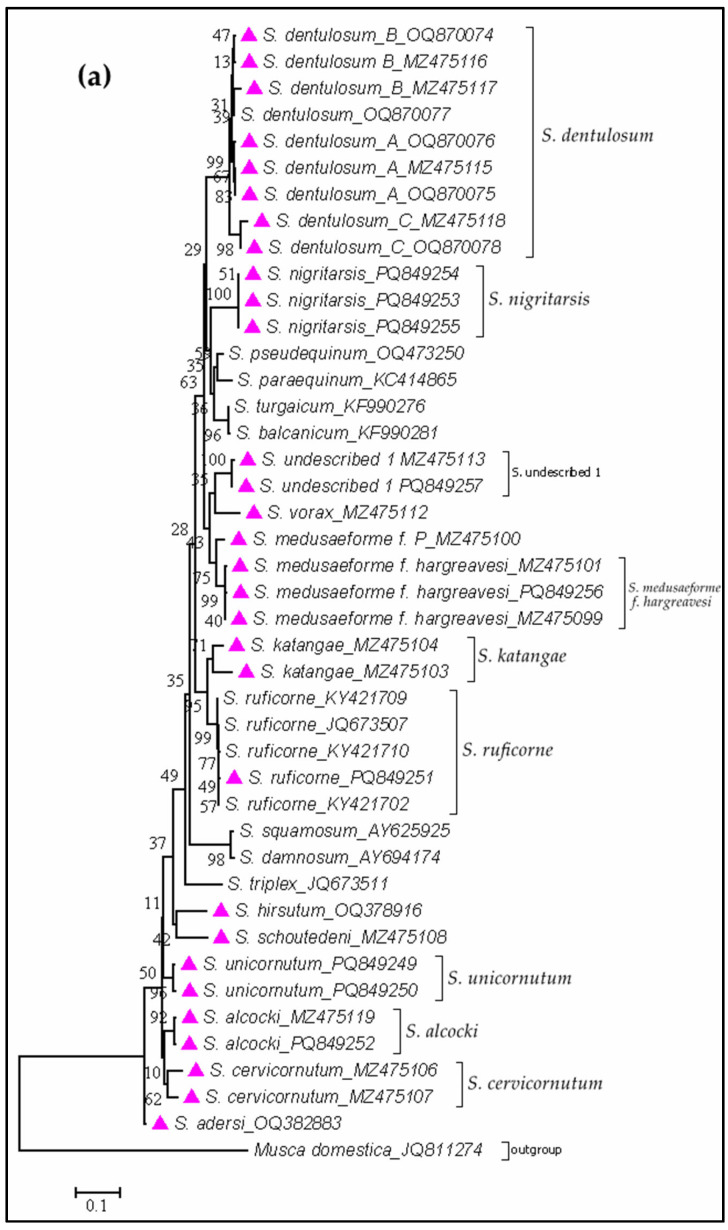
Molecular phylogenetic analysis by neighbor-joining method using ITS2 (**a**) and Cox1 (**b**). Species marked in purple/violet triangles are from this study and others are from the GenBank submitted by other authors (see Appendix A for references) and used in this analysis to align our findings. The tree is drawn to scale, with branch lengths in the same units as those of the evolutionary distances used to infer the phylogenetic tree.

**Table 1 insects-16-00572-t001:** Comprehensive list of Simuliidae species described from Cameroon [13,15].

Subgenus	Species-Group	Species	Authors	Years
BYSSODON Enderlein	Griseicolle	*griseicolle*	Becker	1903
ANASOLEN Enderlein		*dentulosum*	Roubaud	1915
EDWARDSELLUM Enderlein	Damnosum	*damnosum complex*	Theobald	1903
		*damnosum s. str*	Dunbar and Vajime	1981
		*mengense*	Vajime and Dunbar	1979
		*sirbanum*	Vajime and Dunbar	1975
		*squamosum* (*complex*)	Enderlein	1921
		*yahense*	Vajime and Dunbar	1975
		*soubrense*	Ayissi et al.	2022
LEWISELLUM Crosskey		*atyophilum*	Lewis and Disney	1969
		*ovazzae*	Grenier and Mouchet	1959
MEILLONIELLUM Rubtsov		*adersi*	Pomeroy	1922
		*hirsutum*	Pomeroy	1922
METOMPHALUS Enderlein	Bovis	*bovis*	De Meillon	1930
		*eouzani*	Germain and Grenier	1970
		*wellmanni*	Roubaud	1906
	Medusaeforme	*akouense*	Fain and Elsen	1973
		*colasbelcouri*	Grenier and Ovazza	1951
		*crosskeyi*	Lewis and Disney	1970
		*futaense*	Garms and Post	1966
		*hargreavesi*	Gibbins	1934
		*medusaeforme s. str.*	Pomeroy	1920
		*ngouense*	Fain and Elsen	1973
		*tondewandouense*	Fain and Elsen	1973
NEVERMANNIA Enderlein	Loutetense	*loutetense*	Grenier and Ovazza	1951
	Ruficorne	*antibrachium*	Fain and Dujardin	1983
		*aureosimile*	Pomeroy	1920
		*ekomei*	Lewis and Disney	1972
		*katangae*	Fain	1951
		*nigritarse*	Coquillett	1901
		*ruficorne*	Macquart	1838
PHORETOMYIA Crosskey		*afronuri*	Lewis and Disney	1970
		*dukei*	Lewis, Disney, and Crosskey	1969
		*berneri*	Freeman	1954
		*kumboense*	Grenier, Germain, and Mouchet	1966
		*baetiphilum*	Lewis and Disney	1972
		*lumbwanum*	De Meillon	1944
		*rickenbachi*	Germain, Grenier, and Mouchet	1966
POMEROYELLUM Rubtsov	Alcocki	*alcocki*	Pomeroy	1922
		*coalitum*	Pomeroy	1922
		*djallonense*	Roubaud and Grenier	1943
		*duodecimum*	Gibbins	1936
		*vargasi*	Grenier and Rageau	1949
		*garmsi*	Crosskey	1969
		*hissetteum*	Gibbins	1936
		*impukane*	De Meillon	1936
		*johannae*	Wanson	1947
		*oguamai*	Lewis and Disney	1972
	Cervicornutum	*cervicornutum*	Pomeroy	1920
		*leberrei*	Grenier, Germain, and Mouchet	1966
		*palmeri*	Pilaka and Elouard	1999
		*unicornutum*	Pomeroy	1920
	Kenyae	*kenyae*	De Meillon	1940
	Schoutedeni	*audreyae*	Garms and Disney	1974
		*schoutedeni*	Wanson	1947

**Table 2 insects-16-00572-t002:** Sampling points with coordinates, sample collectors, sample type, and species collected in each point.

SN	River	Site	Latitude	Longitude	Date	Collector	Morphological Identification.
1	Vina du Nord	Touboro Vina bridge	7.7502	15.3636	16-March-11	AR	* S. bovis*, * S. cervicornutum*, *S. vorax*
2	Benoue	Near Karna Manga	7.7808	13.5874	05-September-14	AE, DE	*S. kenyae*;
3	Tributary to river Vina du Nord	Aladji Marafat	7.4016	13.5522	10-January-20	PK	* S. nigritarsis*, * S. medusaeforme* f. Pomeroy, * S. adersi*, * S. unicornutum*, *S.* undescribed 2 * S. cervicornutum *
4	Vina du Sud	Vina fall at Galim Pont	7.2100	13.5862	18-June-20	DE	*S. vorax*, *S. nigritarsis*, *S. medusaeforme* f. *hargreavesis*, *S. adersi*
5	Mayo Djouroum	Near Galim	7.2011	13.5930	26-April-19	PK, DE	* S. vorax*, * S. medusaeforme * f. Pomeroy, * S. medusaeforme * f. *hargreavesis*, * S. cervicornutum *
6	Mawong river	Near Befang	6.3242	10.0037	17-November-17	PK	*S. schoutedeni*, *S. unicornutum*, *S. katangae*, *S. hirsutum*, *S. cervicornutum*, *S. medusaeforme* f. *hargreavesis*, *S. alcocki*, *S. dentulosum * *S. ruficorne*, *S. adersi*, *S.* undescribed 1, *S. kenyae*
7	Tunga (Menchum) river	Menchum Falls	6.3069	10.0169	02-August-18	PK	* S. cervicornutum*, *S. unicornutum*, * S. katangae*, * S. dentulosum * * S. hirsutum*, * S. alcocki*, * S. * undescribed 1
8	River near IRAD	Bambui	6.0149	10.2677	29-October-18	PK	* S. cervicornutum*, * S. unicornutum*, * S. katangae*, * S. dentulosum * * S. alcocki*, * S. * undescribed 1
9	Tributary of river Nkam	Mbanga near the slaughterhouse	4.5044	9.5719	28-December-21	PK	*No Simulium* found
10	Lele	Nkongsamba	4.9724	9.9289	29-December-21	PK	* S. cervicornutum*, *S. katangae*, * S. dentulosum * * S. medusaeforme * f. Pomeroy, * S. ruficorne *
11	Boriko	Nkongsamba	4.9544	9.9259	29-December-21	PK	* S. cervicornutum*, * S. katangae*, * S. dentulosum * * S. medusaeforme * f. Pomeroy, * S. ruficorne *
12	Tributary near Total	Nkongsamba	4.9577	9.9300	29-December-21	PK	* S. cervicornutum*, * S. katangae*, * S. dentulosum * * S. medusaeforme * f. Pomeroy, * S. ruficorne *
13	Esoa	Nkongsamba	4.9749	9.9399	29-December-21	PK	* S. cervicornutum*, * S. katangae*, * S. dentulosum * * S. medusaeforme * f. Pomeroy, * S. ruficorne *
14	Vina du Nord	Soramboum	7.7872	15.0061	31-October-16	DE	*S. alcocki*

PK: Pierre Kamtsap; DE: David Ekale; AE: Albert Eisenbarth; AR: Alfons Renz.

**Table 3 insects-16-00572-t003:** Estimates of Average Evolutionary Divergence over Sequence Pairs within Groups by ITS2 region.

Species	Average Divergence	Standard Error
*S. alcocki*	0.005	0.004
*S. dentulosum*	0.024	0.004
*S. nigritarsis*	0	0
*S. hargreavesi*	0.004	0.002
*S. katangae*	0.063	0.013
*S. unicornutum*	0.008	0.005
*S. ruficorne*	0.002	0.001
*S. cervicornutum*	0.053	0.011
**Outgroup**	n/c	n/c

The presence of n/c in the results denotes cases in which it was not possible to estimate evolutionary distances.

**Table 4 insects-16-00572-t004:** Maximum Likelihood Estimate of Substitution Matrix.

From/To	A	T	C	G
**A**	–	4.7139	4.7139	15.5722
**T**	4.7139	–	15.5722	4.7139
**C**	4.7139	15.5722	–	4.7139
**G**	15.5722	4.7139	4.7139	–

## Data Availability

Sequences were deposited in GenBank, and prepared slides are in the Institute for Evolution and Ecology, Department of Comparative Zoology, University of Tübingen, Germany.

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
