# Peer review of "Morphological and Molecular Investigation of Non-Simulium damnosum Black Flies in Cameroon Using Nuclear ITS 2 and Mitochondrial Cox 1 Genes"

_insects, 2025, doi:10.3390/insects16060572_

Round 1
Reviewer 1 Report
Comments and Suggestions for Authors
The paper is properly structured and English is good. The interest in improving black fly species distribution is high and therefore the scientific novelty of the study is evident.
The major issue seems to be the choice to morphologically identify species on the basis of the sole pupae: this can be tricky, without the joint study of adult males. You therefore should declarewhich species could have been more easily misidentified on the basis of your knowledrge of the species group (e.g.: aureum group is nearly impossible to sort on the basis of pupal gills).
About the previous concern, please check the species in figure 2 (n): on the basis of this only picture, that looks more like Simulium (Whilelmia) lineatum than S. pseudequinum (at the basis of the gill there is not cuticule bellows).
All over the paper: please write all species names in italics.
Lines 21-23, in the abstract: the sentence should be supported more consistently, or eliminated.
Lines 37-38 Change: "Larvae and pupae are widespread in fast-flowing water of rivers and tributaries."
Line 38: Some adult females
Line 40: two spaces before Onchocerca
Line 81: when separating words to change line, respect syllables (e.g.: Figu -re)
Line 93: here you can say something about the problems with identification only based on pupae
Line 152: write "cercvicornutum" with small c
Lines 244 -253: the entire concept is well known for black fly species, and sounds a little obvious. With no other information given about the species you are studying, it could be eliminted without loosing anything.
Reviewer 2 Report
Comments and Suggestions for Authors
In the reviewed MS three authors from Germany and Cameroon report on their results of faunistic studies of Simulidae in Cameroon. The authors collected pupae of simulids, investigated their morphology, performed species identification mainly based on differences in gill morphology, obtained partial sequences of 2 gene markers (cox1 + ITS2) and confirmed morphological identification using tree-based molecular approach. The MS has several serious flaws. The Discussion report on fact on ecology and epidemiology that were not tested/reported in the section Results. The authors did not investigate larvi and imago of the simulids, which is important for complete correct ID especially for characterizing the "...poor-studied black fly fauna of Cameroon" (line 78). The phylogenetic analysis reported in the MS should be repeated. The authors are requested to include all possible cox1 and its2 sequences of simulids from GBank in the analysis and show the position of their sequences in the gene trees. The map (fig1) is inaccurate, this map needs revision. Apart from photographs, the authors are requested to provide classical zoological drawings of the simulid morphology as it was done in the papers which they cited in the Section 2.1 [26-28]. This MS could be reorganized in a short illustrated taxonomic list, supplemented with genetic barcodes for each taxon, and submitted to a local journal.
Round 2
Reviewer 2 Report
Comments and Suggestions for Authors
The authors carefully revised the MS according to the remarks by reviewer. The authors are requested to revise all scientific names of simuliids and check if they are in italic. The captions for figures need further revision. "Pink color" is actually violet. It is better to refer to purple/violet triangles. In the Fig.1 green circles provide very poor contrast, it is better to reconsider the color. It is a pity that the authors neglected the remark to provide classical morphological data on simuliids. This fact significantly weakens this MS.
Author Response
We appreciate the reviewer's informative feedback and have carefully considered all proposed modifications (See blue colors).
Scientific Names: We meticulously examined all occurrences of Simuliidae species names in the paper and made sure they were uniformly italicized.
Figure Captions: The captions have been changed to improve clarity. To more appropriately characterize the hue in the figures, we changed "pink color" to "purple/violet.".
Figure 1 Adjustments: We recognize the criticism about the low contrast of the green circles. We modified the color palette to boost visibility and readability.
Classical Morphological Data: We understand the significance of traditional morphological data for simuliid identification. However, our work focuses primarily on molecular and ecological studies, and no traditional morphological data were acquired as part of this research. We highlighted this in the text, emphasizing how our molecular approach supports established taxonomy approaches.